# The Influence of Herbicide Underdosage on the Composition and Diversity of Weeds in Oilseed Rape (*Brassica napus* L. var. *oleifera* D.C.) Mediterranean Fields

**Paola A. Deligios** [1], **Gianluca Carboni** [2], **Roberta Farci** [1], **Stefania Solinas** [1] and **Luigi Ledda** [1,*]

[1] Department of Agriculture, Division of Agronomy and Plant Genetics, University of Sassari, Viale Italia 39, 07100 Sassari, Italy; pdeli@uniss.it (P.A.D.); rfarci@uniss.it (R.F.); ssolinas@uniss.it (S.S.)

[2] Agris Sardegna—Agricultural Research Agency of Sardinia, Viale Trieste 111, 09123 Cagliari, Italy; gcarboni@agrisricerca.it

[*] Correspondence: lledda@uniss.it; Tel.: +39-079-229-230

**Abstract:** Weed flora is considered harmful for crop growth and yield, but it is fundamental for preserving biodiversity in agroecosystems. Two three-year trials were conducted in Italy (two different sites) to assess the effect of six herbicide treatments on the weed flora structure of an oilseed rape crop. We applied metazachlor during the pre-emergence stage at 25%, 50%, 75%, and 100% of the labelled dose (M25, M50, M75, M100); trifluralin (during the first growing season); post-emergence treatment (PE); and a weedy control (W). Species richness, and diversity indices were used to characterize weed flora composition and to evaluate the effect of herbicide treatments on the considered variables. Results highlighted that the weed community is characterized by a higher diversity in underdosed than in M100 treated plots. *Raphanus raphanistrum* and *Sinapis arvensis* were the most common species in M75 and M100 treatments in both sites, while more weed species were detected in underdosed treatments and in weedy plots. The highest Shannon index values were observed in the underdosed treatments. In general, only a slightly similar trend was observed between sites, weed abundance and diversity being positively affected both by low-input herbicide management and by environmental factors (e.g., pedoclimatic situation and previous crop).

**Keywords:** weed abundance; weed richness; Shannon index; low-input weed management

## 1. Introduction

Currently, biodiversity is considered crucial for agricultural landscapes also having a beneficial impact for cropping systems, especially if it does not limit the achievement of optimal yield [1,2]. In recent decades, biodiversity in farmland gradually declined, mainly because of simplified crop rotations, and ever more intensive cultivation practices [3,4]. Consequently, current crop production systems should be rescheduled to limit the dependence on external inputs and at the same time to foster an adequate enlargement of sustainability and thus biodiversity and crop yield [5,6].

Weed flora is considered, to date, one of the main causes that interfere in a relevant way with the quantity and quality of agricultural production, even if, on the other hand, some authors point out that weed flora is also an important element that characterizes the floristic biodiversity of countryside [7,8]. Furthermore, weed flora is fundamental in favoring the biodiversity of a landscape since it offers shelter and nourishment to a wide range of fauna (insects, birds, small mammals) [9–12].

Currently, weed control management scheduling is addressed to limit dependence on herbicides by keeping the weed flora at a tolerable threshold of control instead of maintaining the crop totally free of weeds [13]. The aim is to develop sustainable cropping systems reducing reliance on herbicides by means of lower herbicide doses than labelled ones [14], covering crops or using mulch [15–17], mechanical [18], and flame weeding [19]. From the perspective of herbicide underdosage adoption, some components of weed communities are expected to be altered. Other than weed species density and richness, these components also involved the overall composition and association of weeds [20,21]. The investigation of the phytostructural effects due to herbicide underdosage is a key step in order to set up weed management systems characterized by a reduced reliance on chemical inputs [21].

Serious concerns arose about the adoption of lower herbicide doses than labeled ones, due to the potential onset of herbicide-resistant weed species [22,23]. However, it is strategical to provide herbicides at the right dose and time in order to manage weeds at a tolerable level and to preserve the crop from the phytotoxicity of the herbicide. Previous studies have demonstrated that labelled herbicide doses are recommended to guarantee a successful degree of weed control in different environments characterized by various pedo-climatic characteristics, and weed flora. Moreover, the same manufacturing industries are aware that there is a broad variety of conditions where herbicide use may be kept at reduced doses, although these conditions are somewhat unpredictable [24–26]. Underdosed herbicide use (50–80% reduction with respect to labeled dose) has been reported and applied for maize on over 50% of the maize-cropped area in The Netherlands and over 80% of the maize-cropped in Denmark, German, and France [27]. Additionally, according to Neve et al. [28] the onset of herbicide-resistant weeds is a great threat in monoculture systems with respect to cropping systems where rotation is regularly practiced [29]. Other authors report that crop rotation is one of the determining factors of the soil seedbank and absence of crop rotation may lead to a weed flora assemblage with a lower biodiversity and thus reducing herbicide available options to manage weeds [30,31].

The oilseed rape crop, which we studied in the present paper, is considered a strong rooting-break species and an improver of soil structure [32,33]. For this last reason, in Mediterranean environments, it is commonly grown in an annual rotation with winter cereals [32,34,35], thus minimizing the risk to cause the selection of herbicide-weed resistance when herbicides are applied at reduced doses.

To our knowledge, studies on the effects of underdosage herbicide in oilseed rape are very limited, and, on the same crop, investigations highlighting typical weed composition were mainly carried out on continental environments [36–40].

This study was aimed to analyze the effects of underdosed herbicides and untreated plots on floristic composition and species diversity in a Mediterranean oilseed rape cropping system. Specifically, this analysis was focused on detecting possible changes of the weed flora community in terms of relative weed abundance, richness, and diversity as a result of underdosed herbicide application.

## 2. Materials and Methods

### 2.1. Study Sites

We assessed weed flora composition (structure and assemblage) in oilseed rape fields in two experimental trials established in 2007 at the experimental stations "Mauro Deidda" in Ottava (40°46′ N, 8°29′ E; 81 m a.s.l.) and "San Michele" in Ussana (39°24′ N, 9°05′ E; 114 m a.s.l.) (Tables S1–S3).

Crop genotype, sowing date, density, and row spacing were based on technical recommendations for the region to obtain the highest attainable yield (Table S4) [41–43].

## 2.2. Treatments

Each year, six treatments were tested in a randomized complete block design with three replications (plot size 12 m × 4.5 m) in an Ottava field experiment, and four replications in a Ussana one (plot size 25 m × 6 m).

During the 2007–2008 growing season, metazachlor (Butisan S, 50% a.i.) was distributed at three doses and two application periods: 1000 g a.i. ha$^{-1}$ (M100, labelled dose), 750 g a.i ha$^{-1}$ (M75), 500 g a.i. ha$^{-1}$ (M50), in pre-emergence, and 250 g a.i. ha$^{-1}$ in post-emergence (PE). In the Ussana experiment, the same pre-emergence treatments (M50, M75, and M100) were also tested; only the post-emergence treatment differed since it was applied at a dose of 500 g a.i. ha$^{-1}$ (PE). Moreover, in both sites, a few days before the sowing date, trifluralin (T; Triflene, 48% a.i.) was supplied at the labelled dose (720 g a.i. ha$^{-1}$) (Table 1).

We also considered for both sites a weedy treatment (W), where weeds were untreated throughout the studied growing seasons.

In the following two seasons (Table 1), in place of trifluralin treatment (banned by the European Union because of its toxic effect in water bodies, 2010/455/EU) metazachlor at the rate of 250 g a.i. ha$^{-1}$ was applied before emergence (M25).

During the last growing season, in Ussana site, post-emergence treatment with metazachlor was replaced by a mixture of clopiralid (Lontrel 72 SG, Dow AgroSciences, Bologna, Italy) and propaquizafop (Agil, DuPont, Milan, Italy) (Table 1).

**Table 1.** Summary of treatments applied during the three-year experiment at Ottava (OTV) and Ussana (USN) sites

| Application Time/Treatment | Herbicide Name | Dose (g a.i. ha$^{-1}$) | Acronym | 2008 | | 2009 | | 2010 | |
|---|---|---|---|---|---|---|---|---|---|
| | | | | USN | OTV | USN | OTV | USN | OTV |
| Pre-sowing | Trifluralin | **720** | **T** | x | x | - | - | - | - |
| Pre-emergence | Metazachlor | 250 | **M25** | - | - | x | x | x | x |
| | | 500 | **M50** | x | x | x | x | x | x |
| | | 750 | **M75** | x | x | x | x | x | x |
| | | 1000 | **M100** | x | x | x | x | x | x |
| Post-emergence | | 250 | | - | x | - | x | - | x |
| | | 500 | **PE** | x | - | x | - | - | - |
| | Clopiralid + propaquizafop | 200 + 100 | | - | - | - | - | x | - |
| Weedy | | | **W** | x | x | x | x | x | xd |

## 2.3. Monitoring and Counting

Crop productivity, profitability, and weed dynamic (in terms of total density, total biomass, and weed coverage) were characterized in the same experiments [14]. In the previous study [14] results underlined that the use of low-herbicide doses might ensure a good level of economic return. Concerning the yield level, in both sites reducing by at least a half the labelled dose of herbicide did not result in any significant difference in seed yield with respect to conventional treatment. For the purpose of the present paper, we measured weed frequency and density between the end of March and mid-April, at the full flowering stage of oilseed rape (BBCH code 68). We randomly placed a quadrat (0.25 × 0.25 m) ten times in each plot and we identified and counted species and number of individual plants per species within each quadrat. For the analysis, we grouped species density of the ten quadrats per plot, then we used individual weed species density for indices calculation. We pooled the raw data into two separate databases: "OTV" related to trials carried out in the site of Ottava, and another called "USN" which referred to fields located in Ussana site.

### 2.4. Data Processing and Statistical Analyses

For both sites and experiments, relative weed abundance (Equation (1)) was calculated to determine the presence of weed species in the whole weed flora assemblage [44].

$$Relative\ abundance = \frac{absolute\ abundance\ of\ a\ species}{sum\ of\ all\ absolute\ abundances} \times 100 \tag{1}$$

$$Absolute\ abundance = \frac{total\ number\ of\ individuals\ of\ a\ species}{total\ number\ of\ sampling\ units\ containing\ that\ species} \tag{2}$$

Species richness was the mean number of species in each treatment [36]. The Shannon's diversity index ($H'$), and the Simpson index ($D$) were calculated according to [44–46]:

$$H' = -\sum_{i=1}^{S}(P_i ln(P_i)) \tag{3}$$

$$D = 1 - \sum_{i=1}^{S} P_i^2 \tag{4}$$

where $P_i$ is the proportion of species $i$, and $S$ is the species richness.

Data (http://dx.doi.org/10.17632/gmg62gv9y9.1) were analyzed separately by site because a significant interaction site × year × treatment was observed for most of the variables studied.

Richness, abundance, and diversity indices were analyzed by using the MIXED procedure of statistical software SAS (SAS Institute, Inc., Cary, NC, USA, ver. 9.1). Tukey's multiple comparison test (at $p \leq 0.05$ level) was used to separate means. Block and interaction year by block were considered as random effect, year and herbicide treatment as fixed effects. We found a statistically significant treatment by year interaction for most of the analyzed variables; therefore, data is presented separately by treatment and year.

For a better interpretation of the complex patterns and interactions existing between herbicide treatments and their effects on weed flora we also applied a multivariate statistical approach. The weed flora, in fact, is a multivariate "entity" composed by several variables. Among multivariate approaches the principal component analysis (PCA), which is a widespread ordination method, allows reduction of the complexity of a multivariate dataset with a minimal loss in its informative power [47]. The PCA, in fact, is an effective technique aimed to analyze interrelationships among a large number of variables by transforming a dataset in a reduced number of uncorrelated (orthogonal) variables, called principal components (PC). The first principal component (PC1) is a linear combination of all the variables of the dataset which accounts for the highest variance fraction, while the second principal component (PC2), which is uncorrelated to the first, account for the maximum remaining part and so on [48,49].

In our study the objective of PCA was to assess the effect to the applied herbicide treatments either on crop growth than on weed flora. For both experiments, we applied PCA considering the different weeding control treatments (M100, M75, M50, M25, PE and W) and the variables observed for production, phenology, and the assessments of weed abundance. We first standardized the data matrix, and then we produced biplots deriving from the PCA by using the first two principal components, plotting loadings and scores. Loadings consisted in the chosen variables, while scores corresponded to the treatments applied. The obtained biplot allowed to interpret the relations between the variables and weeding treatments. PCA was performed using the GenStat for Windows 18th edition [50] whereas the graph was obtained by the DBBIPLOT procedure [51].

## 3. Results

### 3.1. OTV Experiment

#### 3.1.1. Weed Richness, Abundance, and Diversity

M25, M50, PE, and W treatments systematically and significantly differed from M75 and M100 in terms of number of identified species throughout the entire duration of the experiment (Figure 1).

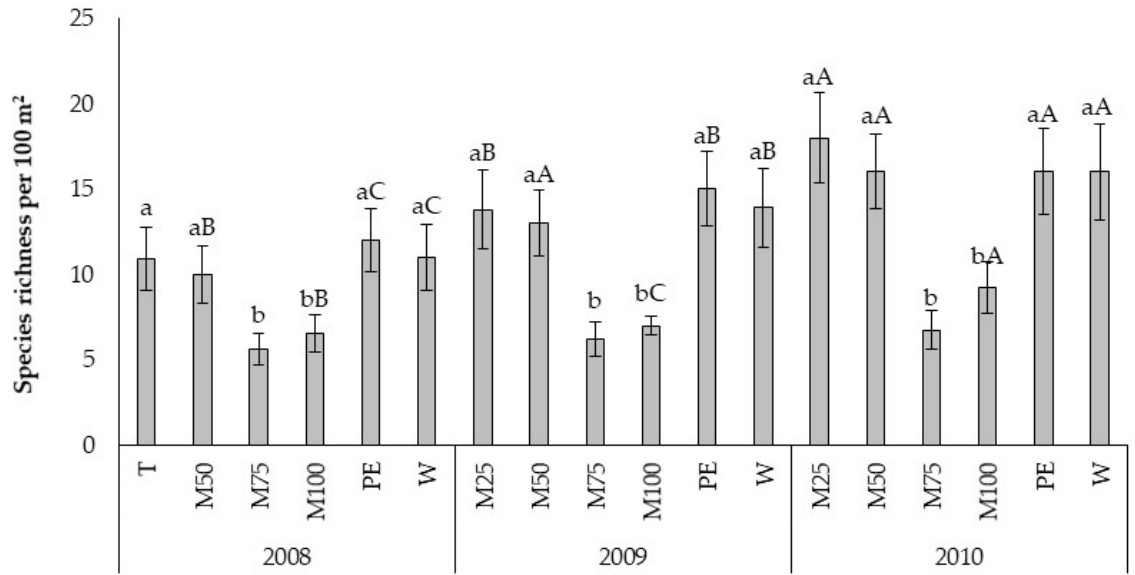

**Figure 1.** Species richness during the three-year experiment at the Ottava (OTV) site. Different letters indicate that means are significantly different among treatments (lower-case letters) and years (upper-case letters) according to Tukey's test (*p* < 0.05). M25, M50, M75, M100 = metazachlor at 25%, 50%, 75%, 100% of the labelled dose, respectively; T = Trifluralin; PE = post-emergence treatment; W = weedy treatment.

In total, throughout the three-year period, the observed species were 23, differently distributed per year and treatment. In particular, nine species were observed only during a growing season, and nine species resulted the most abundant and were observed in all the studied years (Tables 2 and S5).

Fourteen botanical families were observed (Asteraceae, Brassicaceae, Caryophillaceae, Chenopodiaceae, Convolvulaceae, Geraniaceae, Gramineae, Leguminosae, Papaveraceae, Polygonaceae, Primulaceae, Ranunculaceae, Rubiaceae, Scrophulariaceae), the most representative of which were Asteraceae, Brassicaceae, and Gramineae with three species and Chenopodiaceae and Papaveraceae with two species to each family (Tables 2 and S5).

In 2008, in M100 and M75 treatments the most relative abundant species (on average over 15%) were *Raphanus* raphanistrum and *Sinapis arvensis*, *Chrysanthemum coronarium*, and *Papaveraceae* species (*Papaver rhoeas* and *Fumaria officinalis*; Table 2). A similar trend was also observed in the last two growing seasons (2009 and 2010).

**Table 2.** Relative weed abundance (%) during the three-year experiment at the Ottava site. See also data repository for raw data (http://dx.doi.org/10.17632/gmg62gv9y9.1).

| Year | Treatment | | *Anagallis arvensis* L. | *Avena fatua* L. | *Capsella bursa-pastoris* (L.) Med. | *Chenopodium album* L. | *Chris. coronarium* L. | *Convolvulus arvensis* L. | *Fumaria officinalis* L. | *Galium aparine* L. | *Hordeum vulgare* L. | *Lolium rigidum* Gaudin | *Papaver rhoeas* L. | *Ranunculus repens* L. | *Raphanus raphanistrum* L. | *Senecio vulgaris* L. | *Sinapis arvensis* L. | *Sonchus oleraceus* L. | *Vicia sativa* L. |
|---|---|---|---|---|---|---|---|---|---|---|---|---|---|---|---|---|---|---|---|
| 2008 | T | | 0 | 14.5 | 0 | 8.1 | 9.5 | 10.7 | 7.5 | 0 | 10.6 | 8.2 | 7.6 | 0 | 6.9 | 0 | 6 | 10.5 | 0 |
| | M50 | | 0 | 18.1 | 0 | 12.7 | 9.6 | 9.4 | 7 | 0 | 14.4 | 6.7 | 6.7 | 0 | 7.7 | 0 | 7.7 | 0 | 0 |
| | M75 | | 0 | 0 | 0 | 21.4 | 19.9 | 0 | 16.2 | 0 | 0 | 0 | 16.8 | 0 | 13.5 | 0 | 12.1 | 0 | 0 |
| | M100 | | 0 | 0 | 0 | 13.4 | 11.9 | 10.9 | 20.8 | 0 | 0 | 0 | 16.9 | 0 | 15.0 | 0 | 11.1 | 0 | 0 |
| | PE | | 0 | 7.4 | 0 | 3.8 | 10.6 | 1.4 | 10.2 | 8.1 | 11.1 | 9.8 | 8.5 | 0 | 10.1 | 0 | 7.2 | 11.3 | 0 |
| | W | | 0 | 18.9 | 0 | 9.8 | 9.8 | 8 | 6.9 | 0 | 9.7 | 9.4 | 6.7 | 0 | 6.1 | 8.7 | 6.6 | 0 | 0 |
| 2009 | M25 | | 8.2 | 10.6 | 4 | 6.7 | 7.2 | 8.7 | 7.1 | 0 | 0 | 8.9 | 5.5 | 0 | 6.1 | 0 | 6.2 | 0 | 4 |
| | M50 | | 6.1 | 5.6 | 5.6 | 11.9 | 14.1 | 11.3 | 0 | 0 | 0 | 0 | 8.3 | 0 | 7.5 | 0 | 7 | 0 | 1.3 |
| | M75 | | 0 | 0 | 0 | 16.6 | 25.6 | 0 | 14.9 | 0 | 0 | 0 | 17.3 | 0 | 13.5 | 0 | 12.2 | 0 | 0 |
| | M100 | | 0 | 0 | 0 | 15.4 | 7.4 | 22.2 | 14.8 | 0 | 0 | 0 | 12.7 | 0 | 15.5 | 0 | 12 | 0 | 0 |
| | PE | | 4.5 | 7.5 | 7 | 9.9 | 7.4 | 4.2 | 7.9 | 0 | 0 | 9 | 7.2 | 0 | 7.9 | 0 | 7.6 | 6.9 | 2.7 |
| | W | | 10.4 | 10.2 | 9.1 | 7.1 | 6.4 | 7.7 | 6.9 | 0 | 0 | 8.5 | 7.8 | 0 | 7.3 | 0 | 6.3 | 0 | 0.8 |
| 2010 | M25 | | 3.4 | 8.7 | 0 | 7 | 5.5 | 5.7 | 7.9 | 0 | 4.6 | 6.2 | 3.9 | 1.8 | 6 | 5.5 | 5.3 | 0 | 5.6 |
| | M50 | | 8.1 | 9.4 | 0 | 6.3 | 6.2 | 8.9 | 6.7 | 0 | 5.2 | 5.3 | 5.3 | 7 | 6.5 | 5.4 | 5.3 | 0 | 2.2 |
| | M75 | | 0 | 0 | 0 | 15.7 | 15.9 | 0 | 15.6 | 0 | 0 | 13.1 | 14.7 | 0 | 13.1 | 0 | 11.9 | 0 | 0 |
| | M100 | | 0 | 10.3 | 0 | 10.2 | 12.7 | 0 | 13.0 | 0 | 0 | 10.5 | 11.5 | 0 | 12.9 | 0 | 9 | 0 | 9.9 |
| | PE | | 6.6 | 6.8 | 0 | 6.1 | 6.8 | 5.4 | 5 | 0 | 4.7 | 5 | 6.1 | 6 | 5.9 | 5.6 | 6.3 | 0 | 0 |
| | W | | 6.4 | 8.3 | 0 | 6.2 | 5.8 | 6.2 | 5.9 | 0 | 5.1 | 8.6 | 6.4 | 5.1 | 6.6 | 7.6 | 7.3 | 0 | 2.6 |
| | DF | | | | | | | | | *p* > F | | | | | | | | | |
| Analysis of variance | Year (T) | 2 | *** | ns | ** | ns | ns | *** | ns | *** | *** | *** | ns | *** | ns | *** | ns | *** | *** |
| | Treatment (T) | 5 | *** | *** | *** | *** | *** | *** | *** | *** | *** | *** | *** | ** | **** | *** | *** | *** | ** |
| | YxT | 10 | *** | ** | *** | ns | ns | *** | ** | *** | *** | *** | ns | *** | ns | *** | ns | *** | *** |

M25, M50, M75, M100 = metazachlor at 25%, 50%, 75%, 100% of the labelled dose, respectively; T = Trifluralin; PE = post-emergence treatment; W = weedy treatment. The asterisks *, **, ***, or ns indicate statistical differences at *p* < 0.05, *p* < 0.01, *p* < 0.001, or non-significant, respectively.

An opposite pattern was observed in the weedy and in underdosed treatments (M25, M50, and PE). Indeed, due to a higher richness of species, the relative abundance of individual species resulted slightly higher than 10%, and only in a rare case resulted over 15% (*Avena fatua* in W and M25 treatments in 2008) (Table 2).

The species diversity indices varied with herbicide treatments and years, and a significant interaction year by treatment was observed for each index (Table 3). As expected, within each growing season, the highest Shannon's diversity index (H′) was primarily observed in the weedy plots and secondary in the underdosed treated plots (M25, M50, and PE).

Moreover, in line with Shannon index, Simpson index statistically differed among treatments and the lowest values were systematically provided by M75 and M100 treatments. Among growing seasons, 2010 resulted in higher values for each analyzed index and these findings support and confirm the richness ones.

**Table 3.** Shannon ($H'$), and Simpson ($D$) indices of weed flora composition during the three-year experiment in OTV experiment.

| Year | Treatment | Diversity Indices | |
|---|---|---|---|
| | | $H'$ | $D$ |
| 2008 | T | 2.46 b | 0.87 b |
| | M50 | 2.22 cB | 0.89 aB |
| | M75 | 1.71 dB | 0.82 dB |
| | M100 | 1.67 dC | 0.80 eB |
| | PE | 2.27 cC | 0.86 bC |
| | W | 2.72 aB | 0.84 cC |
| 2009 | M25 | 2.48 c | 0.92 a |
| | M50 | 2.19 dB | 0.88 bC |
| | M75 | 1.77 eB | 0.81 dC |
| | M100 | 1.83 eB | 0.81 dB |
| | PE | 2.61 bB | 0.89 bB |
| | W | 2.85 aA | 0.87 cB |
| 2010 | M25 | 2.65 b | 0.93 a |
| | M50 | 2.67 abA | 0.92 aA |
| | M75 | 1.93 dA | 0.85 bA |
| | M100 | 2.08 c | 0.86 bA |
| | PE | 2.79 aA | 0.93 aA |
| | W | 2.72 aB | 0.93 aA |
| | | DF | $p >$ F | $p >$ F |
| Analysis of variance | Year (Y) | 2 | <0.0001 | <0.0001 |
| | Treatment (T) | 5 | <0.0001 | <0.0001 |
| | Y x T | 10 | <0.0001 | <0.0001 |

Different letters indicate means that are significantly different among treatments (lower-case letters) and years (upper-case letters) according to Tukey test ($p < 0.05$). M25, M50, M75, M100 = metazachlor at 25%, 50%, 75%, 100% of the labelled dose, respectively; T = Trifluralin; PE = post-emergence treatment; W = weedy treatment.

### 3.1.2. Principal Component Analysis

The PC1 was characterized by positive values for yield, crop biomass, pod number per plant, days from sowing to emergence and from sowing to flowering and for Brassicaceae species frequency (Table 4). Higher doses of pre-emergence treatments (M100, M75, M50) are positively correlated with these variables (Figure 2). The majority of treatments with lower dose of metazachlor (M25, PE) and the weedy one (W), which have negative values of the first component, are located in opposite position respect to yield, crop biomass, pod number per plant, indicating a negative crop response compared to higher dose of metazachlor (M100, M75, M50) treatments. In addition, the positive values showed by Brassicaceae family underscore that it is probably the most difficult family to control and to which, moreover, the main crop belongs.

**Table 4.** Acronyms, loadings, eigenvalues, and variances accounted by the first two PCs on the means of 11 variables considered.

| Variable | Acronym | Principal Component | |
|---|---|---|---|
| | | PC1 | PC2 |
| Pod number (n.) | Pod_N | 0.290 | 0.052 |
| Crop biomass (kg ha$^{-1}$) | Crop_B | 0.319 | −0.067 |
| Yield (kg ha$^{-1}$) | Yield | 0.311 | 0.193 |
| Weed coverage (%) | Weed_Cov % | −0.313 | 0.078 |
| Sowing-emergence (days) | S-Em | 0.290 | 0.465 |
| Sowing-flowering (days) | S-Fl | 0.275 | 0.543 |
| Brassicaceae (%) | Brass % | 0.304 | 0.154 |
| Graminae (%) | Gram % | −0.301 | 0.287 |
| Papaveraceae (%) | Pap % | −0.316 | 0.151 |
| Other weeds (%) | Oth % | −0.308 | 0.297 |
| Weed biomass (kg ha$^{-1}$) | Weed_B | −0.286 | 0.470 |
| Eigenvalue | | 9.561 | 0.833 |
| % Variance | | 86.9 | 7.6 |
| % Accumulated variance | | 86.9 | 95.4 |

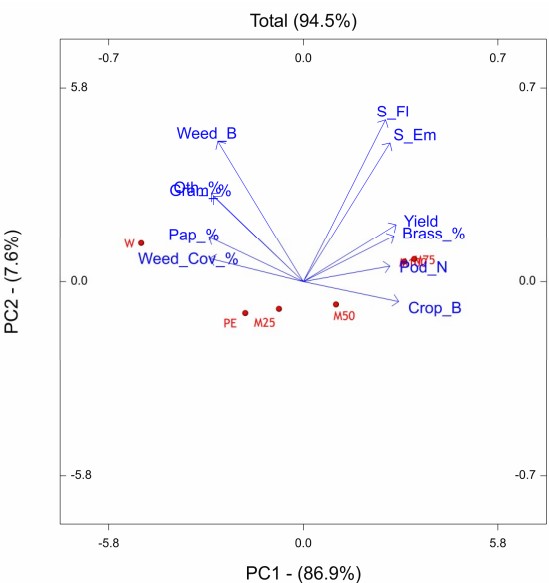

**Figure 2.** Principal component analysis (PCA) biplot showing patterns and interactions between crop variables and weeds with herbicide treatments (red labels): M100, labelled metazachlor dose; M75, 75% of the labelled metazachlor dose; M50, half of labelled metazachlor dose; M25, 25% of labelled metazachlor dose; W, weedy plot. Acronyms used: Yield, yield (t ha$^{-1}$); Pod_N, pod number per plant (n.); Crop_B, crop biomass at harvest (kg ha$^{-1}$); S-Em, sowing-emergence (days); S-Fl, sowing-flowering (days); Weed_Cov, weed coverage (%); Brass %, Brassicaceae; Pap %, Papaveraceae (%); Gram %, Graminae (%); Oth %, other weed families (%). The percentage of the total variance accounted for by the two first components is shown in the corresponding axes.

*3.2. USN Experiment*

3.2.1. Weed Richness, Abundance, and Diversity

Species richness in M100 and M75 was statistically different from the underdosed (M25, M50, and PE) and weedy treatments in all three years (Figure 3). Moreover, species richness differed among years, and the lowest number of species was recorded during the second growing season (Figure 3).

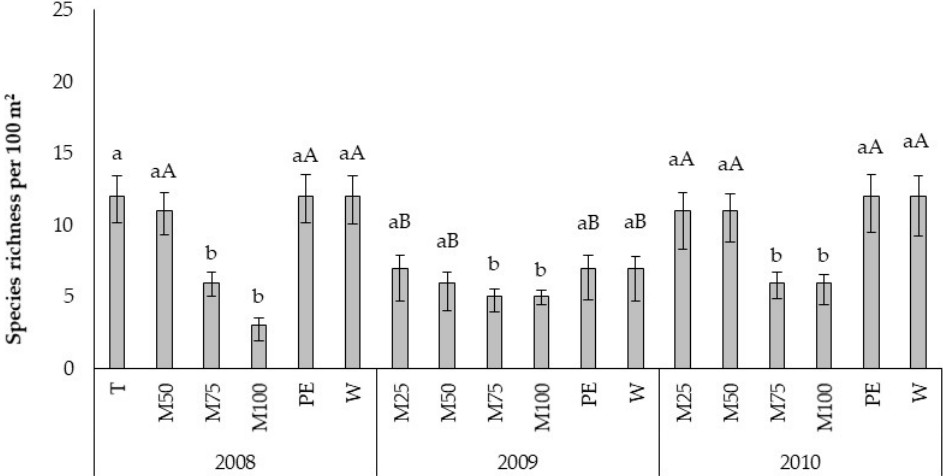

**Figure 3.** Species richness during the three-year experiment in the Ussana (USN) experiment. Different letters indicate means that are significantly different among treatments (lower-case letters) and years (upper-case letters) according to Tukey's test ($p < 0.05$). M25, M50, M75, M100 = metazachlor at 25%, 50%, 75%, 100% of the labelled dose, respectively; T = Trifluralin; PE = post-emergence treatment; W = weedy treatment.

The relative abundance of individual species differed among years and treatments. Table 5 shows the presence of 17 weed species (14 dicotyledons and three monocotyledons) that belong to 10 botanical families. The most represented families were *Leguminoseae* and *Gramineae* with three species, and *Asteraceae*, *Brassicaceae*, and *Papaveraceae* with two species per family. In 2008, the relative abundance of the two *Papaveraceae* species (*Papaver rhoeas* and *Fumaria officinalis*), *Medicago polymorpha*, and *Sinapis arvensis* with regard to M75 and M100 treatments was over 20%. The same scheme was also observed in 2010 and in 2009 with *Raphanus raphanistrum* in place of *Sinapis arvensis* (Table 5). In underdosed treatments and in weedy plots relative abundance of each individual species was slightly lower because of the higher species richness (Table 5). Indeed, in the labelled treated plots, four species (*R. raphanistrum*, *M. polymorpha*, *F. officinalis* and *P. rhoeas*) became dominant within the community, whereas the dominant species in underdosed treatments showed a similar relative abundance (Table 5).

**Table 5.** Relative weed abundance (%) during the three-year experiment at Ussana site. See also data repository for raw data (http://dx.doi.org/10.17632/gmg62gv9y9.1).

| Year | Treatment | | *Avena fatua* L. | *Calendula officinalis* L. | *Chenopodium album* L. | *Ecballium elaterium* (L.) A. Rich. | *Fumaria officinalis* L. | *Lolium rigidum* Gaudin | *Medicago polymorpha* L. | *Papaver rhoeas* L. | *Phalaris* sp. pl. | *Plantago lanceolata* L. | *Polygonum aiculare* | *Ranunculus repens* L. | *Raphanus raphanistrum* L. | *Silybum marianum* (L.) Gaertn. | *Sinapis arvensis* L. | *Trifolium* sp. pl. | *Vicia sativa* L. |
|---|---|---|---|---|---|---|---|---|---|---|---|---|---|---|---|---|---|---|---|
| 2008 | T | | 10.6 | 10.7 | 12.1 | 0 | 9.2 | 7.8 | 7.8 | 8.6 | 0 | 0 | 6.2 | 0 | 1.9 | 7.5 | 7.8 | 9.9 | 0 |
| | M50 | | 9.5 | 17.4 | 12.4 | 0 | 11 | 7.8 | 10.5 | 8.7 | 0 | 0 | 2 | 0 | 2.2 | 10.4 | 8.1 | 0 | 0 |
| | M75 | | 0 | 0 | 4.7 | 0 | 21.8 | 0 | 22.7 | 24.7 | 0 | 0 | 0 | 0 | 5.2 | 0 | 20.8 | 0 | 0 |
| | M100 | | 0 | 0 | 0 | 0 | 38.3 | 0 | 53.8 | 0 | 0 | 0 | 0 | 0 | 8.0 | 0 | 0 | 0 | 0 |
| | PE | | 7.6 | 9.2 | 8.9 | 0 | 8.7 | 8 | 8.9 | 8.4 | 0 | 0 | 5.6 | 0 | 9.2 | 0 | 9.1 | 9.3 | 7.2 |
| | W | | 8.5 | 7.7 | 8.1 | 0 | 7.3 | 8.9 | 8.1 | 8 | 12 | 0 | 14.6 | 0 | 2 | 7.7 | 7.3 | 0 | 0 |
| 2009 | M25 | | 0 | 0 | 0 | 0 | 15.5 | 14.9 | 13.5 | 13.8 | 15.1 | 0 | 0 | 0 | 14.1 | 0 | 13.2 | 0 | 0 |
| | M50 | | 0 | 0 | 0 | 0 | 19.2 | 0 | 18.8 | 17.5 | 10.9 | 0 | 0 | 0 | 17.9 | 0 | 15.7 | 0 | 0 |
| | M75 | | 0 | 0 | 0 | 0 | 21 | 0 | 23.1 | 25.5 | 0 | 0 | 0 | 0 | 24.7 | 0 | 5.6 | 0 | 0 |
| | M100 | | 0 | 0 | 0 | 0 | 20.4 | 0 | 37.9 | 14.9 | 0 | 0 | 0 | 0 | 16.5 | 0 | 10.3 | 0 | 0 |
| | PE | | 13.2 | 0 | 0 | 0 | 13.8 | 14.1 | 19.4 | 14.5 | 0 | 0 | 0 | 0 | 15.4 | 0 | 9.6 | 0 | 0 |
| | W | | 0 | 0 | 0 | 0 | 13.8 | 14.6 | 13.1 | 14.6 | 15 | 0 | 0 | 0 | 14.2 | 0 | 14.6 | 0 | 0 |
| 2010 | M25 | | 0 | 0 | 0 | 9.8 | 10.5 | 10.5 | 9.8 | 8.5 | 10 | 11.4 | 0 | 10.7 | 9.5 | 8.3 | 0 | 0 | 1 |
| | M50 | | 5.5 | 0 | 0 | 11.3 | 9.6 | 10.3 | 10.6 | 10.7 | 11 | 9.8 | 0 | 9.8 | 8.6 | 0 | 2.9 | 0 | 0 |
| | M75 | | 0 | 0 | 0 | 0 | 19.3 | 9.3 | 18.3 | 17.8 | 0 | 0 | 0 | 0 | 17.3 | 0 | 18 | 0 | 0 |
| | M100 | | 0 | 0 | 0 | 0 | 25.2 | 9.5 | 14.4 | 14.9 | 0 | 0 | 0 | 0 | 29.4 | 0 | 6.6 | 0 | 0 |
| | PE | | 0 | 0 | 0 | 9.7 | 11.4 | 9.5 | 9.3 | 9 | 8.8 | 8.3 | 0 | 2.5 | 6.3 | 8.4 | 8.9 | 0 | 7.9 |
| | W | | 6.9 | 0 | 0 | 7.4 | 8.5 | 7.2 | 8.7 | 8.5 | 8.6 | 9.5 | 0 | 9 | 8.5 | 9.3 | 7.9 | 0 | 0 |
| | DF | | | | | | | | | *p* > F | | | | | | | | | |
| Analysis of variance | Year (T) | 2 | *** | *** | *** | *** | ns | ** | * | *** | *** | *** | ** | *** | * | *** | * | *** | ** |
| | Treatment (T) | 5 | *** | *** | ** | *** | *** | *** | *** | *** | *** | *** | *** | ** | *** | *** | ** | *** | *** |
| | YxT | 10 | *** | *** | *** | *** | * | *** | * | * | *** | *** | *** | *** | ** | *** | *** | *** | ** |

M25, M50, M75, M100 = metazachlor at 25%, 50%, 75%, 100% of the labelled dose, respectively; PE = post-emergence treatment; W = weedy treatment. The asterisks *, **, ***, or ns indicate statistical differences at $p < 0.05$, $p < 0.01$, $p < 0.001$, or non-significant, respectively.

In all studied years, there was a statistical difference between labelled dose treatment and the underdosed ones with regard to Shannon index. Diversity was similar among the three underdosed treatments (both in pre-emergence and post-emergence) and in weedy plot (Table 6).

**Table 6.** Shannon ($H'$), and Simpson ($D$) indices of weed flora composition during the three-year experiment in the USN experiment.

| Year | Treatment | Diversity Indices | |
|---|---|---|---|
| | | $H'$ | $D$ |
| 2008 | T | 2.07 b | 0.91 a |
| | M50 | 1.99 bA | 0.90 a |
| | M75 | 1.32 cB | 0.87 a |
| | M100 | 0.78 dB | 0.63 b |
| | PE | 2.38 aA | 0.89 a |
| | W | 2.52 aA | 0.88 a |
| 2009 | M25 | 1.78 a | 0.88 |
| | M50 | 1.7 aB | 0.80 |
| | M75 | 1.4 abB | 0.74 |
| | M100 | 1.12 bA | 0.60 |
| | PE | 1.82 aB | 0.85 |
| | W | 1.71 aB | 0.90 |
| 2010 | M25 | 2.08 a | 0.92 a |
| | M50 | 2.18 aA | 0.88 a |
| | M75 | 1.68 bA | 0.81 ab |
| | M100 | 1.43 bA | 0.74 b |
| | PE | 2.33 aA | 0.89 a |
| | W | 2.52 aA | 0.90 a |
| | | DF | $p > F$ | $p > F$ |
| | Year (Y) | 2 | 0.002 | 0.0318 |
| Analysis of variance | Treatment (T) | 5 | <0.0001 | <0.0001 |
| | Y x T | 10 | 0.0008 | 0.6655 |

Different letters indicate means that are significantly different among treatments (lower-case letters) and years (upper-case letters) according to Tukey test ($p < 0.05$). M25, M50, M75, M100 = metazachlor at 25%, 50%, 75%, 100% of the labelled dose, respectively; T = Trifluralin; PE = post-emergence treatment; W = weedy treatment.

### 3.2.2. Principal Component Analysis

The first two PCs (Figure 4 and Table 7) explained more than 88% of the total variability of the data, which is a sufficient amount to interpret the complex interactions existing between weed treatments and their effects on the crop. The metazachlor pre-emergence treatments (M100, M75, M50, M25) are almost arranged along the PC1, which accounts for more than 70% of the entire variability, and according to a decreasing dose gradient from the conventional (M100) to the control treatment. The production variables (yield, seed weight, and oil content) are grouped in the right part of the biplot and are positively correlated (small angle between vectors) with the higher doses (M100, M75, M50). Therefore, yield, seed weight, and seed oil content are favored by increasing doses of metazachlor in pre-emergence treatments. This is justified by the length of the vectors: the greatest treatment effect corresponds to the longest vector. Doses included between the full and the half ones distributed before the crop emergence (right part of the biplot) allow to reduce the weed relative abundance (left part) which is greater in the weedy, PE, and M25 treatments. The beneficial effects of higher doses of pre-emergence treatments (M100, M75, M50) are counterbalanced by the lengthening of the phase between sowing and emergence and sowing-flowering.

This effect, along with the tendency of plants to be lower with the higher doses distributed in pre-emergence (M100, M75, M50), show the presence of an increasing phytotoxic effect of metazachlor due to an increase in applied doses. Indeed, plant height vector opposite to M100, M75, M50 vectors highlight a negative correlation (the angle is close to 180°).

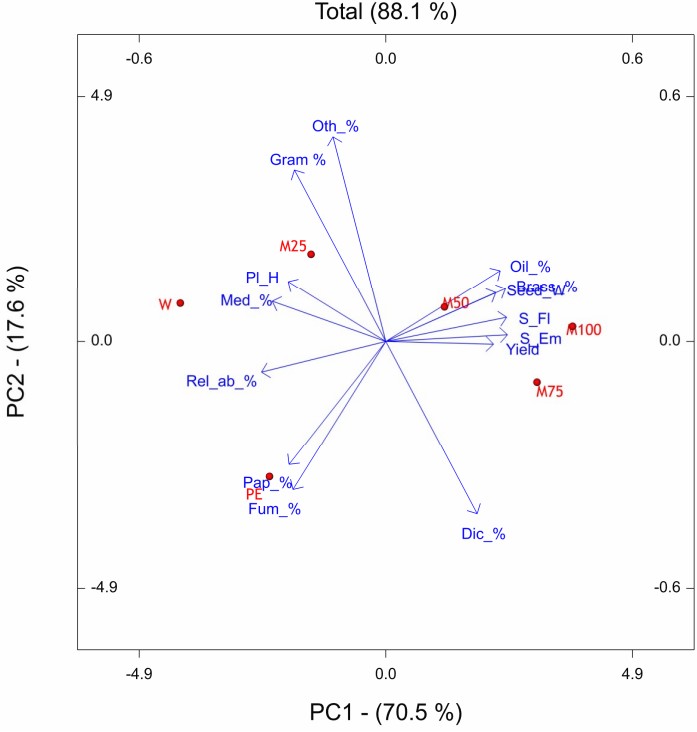

**Figure 4.** PCA biplot showing patterns and interaction between crop variables and weeds with herbicide treatments (red labels): M100, labelled metazachlor dose; M75, 75% of the labelled metazachlor dose; M50, half of the labelled metazachlor dose; M25, 25% of labelled metazachlor dose; W, weedy plot. Acronyms used: Yield, yield (t ha$^{-1}$); Seed_W, seed weight (mg); Oil %, seed oil (%); P H, plant height (cm); S-Em, sowing-emergence (days); S-Fl, sowing-flowering (days); Weed Ab, weed abundance (%); Dic %, dicotyledons (%); Mon %, monocotyledons (%); Brass %, Brassicaceae; Fum %, Fumaria officinalis (%); Pap %, Papaver rhoeas (%); Med %, Medicago polymorpha %); Oth %, other weeds. The percentage of the total variance accounted for by the two first components is shown in the corresponding axes.

**Table 7.** Acronyms, loadings, eigenvalues, and variances accounted by the first two PCs on the means of 14 considered variables.

| Variable | Acronym | Principal Component | |
|---|---|---|---|
| | | PC1 | PC2 |
| Yield (t ha$^{-1}$) | Yield | 0.273 | −0.007 |
| Seed weight (mg) | Seed W | 0.279 | 0.124 |
| Seed Oil (%) | Oil % | 0.290 | 0.180 |
| Plant height (cm) | P H | −0.247 | 0.152 |
| Sowing-emergence (days) | S-Em | 0.308 | 0.017 |
| Sowing-flowering (days) | S-Fl | 0.306 | 0.061 |
| Weed cover (%) | Weed Cov % | −0.314 | −0.077 |
| Dicotyledons (%) | Dic % | 0.231 | −0.437 |
| Monocotyledons (%) | Mon % | −0.231 | 0.437 |
| Brassicaceae (%) | Brass % | 0.303 | 0.134 |
| Fumaria (%) | Fum % | −0.235 | −0.375 |
| Papaver (%) | Pap % | −0.245 | −0.311 |
| Medicago (%) | Med % | −0.288 | 0.101 |
| Other weeds (%) | Oth % | −0.134 | 0.521 |
| Eigenvalue | | 9.869 | 2.460 |
| % Variance | | 70.5 | 17.6 |
| % Accumulated variance | | 70.5 | 88.1 |

## 4. Discussion

Sustainable intensification, assuming a decrease in the use of agrochemicals (fertilizer, herbicide etc.), has the positive aspect to promote ecosystem services within which the conservation of weed flora diversity occurs without a reduction of arable land for food production [52,53]. The primary goal of our study was to determine whether reduction of herbicide dose used would significantly increase weed diversity and composition. Our study represents an assessment of the weed vegetation of oilseed rape fields in a Mediterranean environment, aiming to highlight how weed composition is affected by an underdose herbicide factor. In our experiment, weeding management based upon the non- and low use of herbicides fostered a significantly greater weed species diversity than did conventional management, namely with the labelled herbicide dose. The magnitude of change in weed community composition observed in this study was slightly smaller than that observed by Fried et al. [37] in France and by Hanzlik and Gerowitt [38] in Germany. However, our findings showed a slightly higher diversity than studies carried out in Denmark and United Kingdom in oilseed rape arable fields [40,54].

In particular, in the USN experiment, environmental and site effects played a key role in highlighting specific patterns in weed composition, as, being equal treatments, the site of Ussana recorded each year a number of species lower than Ottava. The importance of landscape heterogeneity (e.g., pedo-climatic characteristics, previous land-uses) in the arrangement of a weed community was previously highlighted by other authors [55], and sometimes, contrasting results were reported, likely as a consequence of weed assembly variation rather than consistent shifts in community composition [56–58]. As expected, the weed species richness and diversity increased in the untreated control [59]. Only six weed species (*Avena fatua*, *Fumaria officinalis*, *Lolium rigidum*, *Papaver rhoeas*, *Raphanus raphanistrum*, and *Sinapis arvensis*), out of all observed species, were dominant being recorded every year in every site. Among these, only two grass weeds were highly abundant (*Avena fatua* and *Lolium rigidum*), and the relative abundance of these two species was higher in underdosed treatments. This last result is also supported by other studies that found high grass weed count in low-input management systems and high broad-leaf count in conventional managed system [58]. Specifically, with respect to other studies focused on oilseed rape weed flora (even if in a different environment), some species resulted common for oilseed rape independently from location, underlining that those weeds are oilseed rape crop-specific [37,39]. The latter include species such as *Galium aparine*; *Convolvolus arvensis*; *Papaver rhoeas*; *Senecio vulgaris*; *Stellaria media*; *Capsella bursa-pastoris*; *Fumaria officinalis*; *Raphanus raphanistrum*, and species belonging to genus *Chenopodium*, *Polygonum*, *Geranium*, *Veronica*, *Sonchus*, and *Vicia*. By contrast, some other species are not typical of oilseed rape crop weed flora community, but in any case some of them (e.g., *Anagallis arvensis*, *Chrisanthemum coronarium*, *Medicago* sp. pl., *Ranunculus* sp. pl., *Plantago* sp.pl., *Sylibum marianum*, *Trifolium* sp. pl.) are included as weed common species of arable fields in the Mediterranean region [60–62]. In particular, *Chrysanthemum coronarium* and *Sylibum marianum* are considered common weeds species of cereal crops of the Mediterranean basin [63–66], thus, suggesting the importance of previous crop and rotation in developing the weed flora community of a species [67]. This assertion is also supported by other studies who stated that crop rotation influences weed species [68] as well as weed seed banks [69]. Moreover, Koocheki et al. [70] pointed out that different rotations that include crops with different life cycles such as winter wheat-maize and winter wheat-sugar beet could lead to additional benefits in reducing the weed seed bank and the incidence of perennial grasses and broadleaf species.

Less common species were *Beta vulgaris*, *Calendula officinalis*, *Ecballium elaterium*, and *Plantago lanceolata.* Environmental conditions greatly influenced the occurrence of some weeds. The observed differences in species frequency and diversity of weed flora assemblage among years might be attributed to the impact of environmental factors such as rainfall. Some authors [71,72], indeed, reported that in addition to nutrients (N), weeds growth cycle is affected by water availability. In our findings, *Ranunculus repens* was observed only during the last growing season (2010) and in both sites. The spring of the last growing season was characterized by a cumulated rainfall higher than the long-term series typical for the region (Tables S1 and S2), and this fostered the occurrence of *R. repens*

which is a weed species mostly found during wet spring [73] and in low-input herbicide management system [74]. Other authors stated that at least 10 mm of rainfall is required for the emergence of all the species, with significantly higher germination rates at rainfall amounts of >20 mm [75] and with cumulated rainfall over consecutive days, rather than single rainfall events of the same amount. The length of time that the soil surface remained above the permanent wilting point, allowing the imbibition of water, was also an important factor [75]. Furthermore, in our study *C. album* was observed at the Ottava site in each growing season, whereas, at the Ussana site, it failed to be found in 2009 and in 2010. Grundy et al. [76] found a relationship between the winter mean temperatures and the relative intensity of *C. album* infestation observed across various environments. The latter ascribed the flush of emergence of *C. album* to the cumulated time spent below some critical temperature greater than 0 °C since a deeper winter chilling may have a greater dormancy breaking effect than a relatively mild winter chill. This last explanation is consistent with the results achieved in our study and with the climatic conditions of the sites under study. Indeed, Ottava site, during the period under study, was characterized by a minimum temperature at least 2–3 °C lower than Ussana site during winter season (Tables S1 and S2). The decline in weed flora composition and diversity caused by herbicide treatments is confirmed by different studies. Edesi et al. [77] reported a downward trend of Shannon's index due to herbicide use. Other previous studies [74,78] supported our findings reporting the highest species diversity in low-input system and at untreated plots level. It is noteworthy to mention that different diversity patterns concerning treatments among sites suggests that different agronomic practices and environmental factors may interact in a complex way with treatments and affect the weed diversity within communities [79]. According with findings of previous studies [80–82], it seems that weed communities differed first among sites, while weed shifts within each site is mainly associated with growing season and herbicide treatments.

The PCs of OTV site reveals that the underdosed treatments PE, M25, and M50 resulted shifted downward with respect to W, M100, and M75. The weed flora related traits were mainly associated with W treatment, while, the crop related traits were associated to M75 and M100. The period from sowing to emergence and sowing to flowering was most sensitive to M75 and M100, indeed the number of days to emerge and to flower resulted higher with respect to the other treatments. We observed the same finding also in the site of Ussana, where higher doses of pre-emergence treatments (M100, M75, M50) caused phytotoxic effects on oilseed rape by lengthening the phase between sowing and emergence and sowing-flowering and reducing crop growth. The phytoxicity of metazachlor on the oilseed rape crop when applied at labelled dose was also reported by Vercampt et al. [83,84].

As a result of application of labelled dose, the metazachlor sensitive species were strongly restrained and thus the community consists of fewer species mainly belonging to Brassicaceae botanical family. If M100, M75, M50 treatments are effective for the general control of weeds, these favor, in proportion, the presence of weeds belonging to the Brassicaceae family (Brassicaceae % vector with smaller angle with M100, M75, M50 vectors). This effect is due to the fact that with metazachlor, which is effective both with monocotyledons and dicotyledons, results are less effective with the physiologically closer weeds such as those of the Brassicaceae family. Some of these findings are consistent with results reported by other authors who claimed that in oilseed rape intensively cropped, a shift in weed vegetation occurred by favoring the Brassicaceae weed species [39] and suggesting that herbicide pressure may lead to a genetically closer community structure [85]. This behaviour at community level might arise from the phenomenon known as 'crop mimicry' [86], in which each individual of a population is selected on the basis of its morphological or bio-chemical similarity to the crop enabling them to escape some selection pressures. An increase in the abundance of tolerant and perennial species is not desirable because they are more harmful than annuals. Therefore, weed species diversity should be sought, as the interspecific competition of weeds prohibits one species from becoming a dominant, 'problem' weed. Finally, that time of herbicide application (pre- vs. post-emergence application) influenced weed flora assemblage as observed in previous studies, even though they are not considered as the main cause of weed community variation [87].

## 5. Conclusions

In this study, we showed the positive aspects derived by a reduced herbicide dose use on weed diversity maintenance. However, as indicated by the site-specific results, the extent of such positive advantages vary depending on growing season meteorological trend, soil properties, and/or previous land-uses. Therefore, building on the results of our study, additional experiments should establish which kind of agricultural management and practice may ensure the highest advantage to weed community of oilseed rape cultivation.

**Supplementary Materials:** The following is available at http://www.mdpi.com/2071-1050/11/6/1653/s1. Table S1: Cumulative precipitation and mean temperature during 2007–2010 period and the long-term meteorological series (1973–2010) at the site of Ussana (39° N, 9° E; 114 m a.s.l.), Table S2: Cumulative precipitation and mean temperature during 2007-2010 period and the long-term meteorological series (1958–2008) at the site of Ottava (40° N, 8° E; 81 m a.s.l.), Table S3: Soil properties at Ussana (Petrocalcic Palexeralf) [88] and Ottava (Lithic Xerorthents) experimental sites at the beginning of experiment in 2007, Table S4: Agronomical management at both experimental fields, Table S5: Less relative weed abundant (%) species during the 3-year experiment at Ottava site. See also data repository for raw data (http://dx.doi.org/10.17632/gmg62gv9y9.1).

**Author Contributions:** P.A.D. conducted the experimental work, analysed the data and wrote the paper; G.C. conducted the experimental work, conceived and designed the field experiment, supported experiment costs, analysed the data and wrote the paper; R.F. conducted the experimental work; S.S. revised and edited the paper; L.L. (corresponding author) conceived and designed the field experiment, supported experiment costs, analysed the data and wrote the paper.

**Funding:** This work was jointly supported by the Italian Ministry of Agricultural, Food and Forestry Policies ("Bioenergie" project); and by the Sardinia Region ("Biocarburanti" project).

**Acknowledgments:** The authors are grateful to the technicians of the experimental farms for their technical support.

**Conflicts of Interest:** The authors declare no conflict of interest.

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
