# Peer review of "The Influence of Herbicide Underdosage on the Composition and Diversity of Weeds in Oilseed Rape (Brassica napus L. var. oleifera D.C.) Mediterranean Fields"

_sustainability, doi:10.3390/su11061653_

Round 1

Reviewer 1 Report

A good 3 year study, which makes this work important.

Generally quite well written, however, I ave some minor suggestions for authors.

Line 196: If values can be >1, then Pielor Index cannot range between 0 - 1.

Line 286 rewrite "Permit" wrong word

Line 312 add an by "an" underdose

Line 333/334 remove "and a reference herein in bracket [54]

Line 335 add "a" different, remove field after ripe

Line 339/340 remove "encompass as" not needed

Line 342 change field to fields in "the" Mediterranean

Line 344 of "the" Mediterranean basin

Line 370 results"are" less

Line 372 cropped "," add comma

Line 373 indicating "," add comma

I think the value of crop rotation using a different genus which varies herbicides and cultivation to prevent build ups of perennial weed species could be added to the discussion.

Author Response

Response to Reviewer#1

A good 3 year study, which makes this work important.

Generally quite well written, however, I ave some minor suggestions for authors.

Line 196: If values can be >1, then Pielor Index cannot range between 0 - 1.

Thank you for this advice. We meant that in the majority of studies/cases Pielou Index assumes values between 0 and 1, when weed flora diversity is relevant the value is higher than 1. We deleted the sentence to avoid misunderstanding and according also with other reviewers’ suggestions (LL. 227-229).

Line 286 rewrite "Permit" wrong word

Thank you for this advice, "permit" has been replaced by "allow" (L. 321).

Line 312 add an by "an" underdose

"an" before "underdose" has been added (L. 347).

Line 333/334 remove "and a reference herein in bracket [54]

Done, "and a reference herein in bracket [54]" has been deleted (L. 369).

Line 335 add "a" different, remove field after ripe

Done, "a" has been added, and "field" has been removed (L. 371).

Line 339/340 remove "encompass as" not needed

Done, "encompass as" has been deleted (L. 375).

Line 342 change field to fields in "the" Mediterranean

Done, "field" has been changed to "fields", and "the" has been added (L. 378).

Line 344 of "the" Mediterranean basin

Done, "the" has been added (L. 380).

Line 370 results"are" less

Done, "are" has been added (L. 441).

Line 372 cropped "," add comma

Done, the comma has been added (L. 444).

Line 373 indicating "," add comma

Done, the comma has been added (L. 445).

I think the value of crop rotation using a different genus which varies herbicides and cultivation to prevent build ups of perennial weed species could be added to the discussion.

Thank you for this suggestion, we added a new paragraph to discuss this issue (LL. 381-386).

Reviewer 2 Report

Review on sustainability-453011 “The influence of herbicide underdosage on the composition and diversity of weeds in oilseed rape Mediterranean fields”

This manuscript evaluates the effects of different herbicide treatment levels on weed floral composition of oilseed rape (Brassica napus L.?? scientific name should be included) at two different sites in Italy. I think the method used by the authors is vaguely enough, but the statistical analysis part is not clear and strong. The power of their work could have been stronger if they had more sites than just two (particularly because of the high variation between sites), but perhaps the authors could address this in the methods or in the discussion, including some logistical constraints. The manuscript is not clearly written at many places and needs to be improved to justify for the publication. Also, language in several places could have been more straightforward (i.e., active voice instead of passive) and concise. For example, there is no need of explaining something like “The results of the PCA are shown in table 7 (L275)…”. Please pay special attention to maintain the standards of scientific writings. Overall, the results of this study are interesting and useful. Detailed comments are below.

Detailed comments.

Abstract:

L13, Influence is not the right word here .

L16, authors need to pay attention to the sentence structure. Also, need “and” after PE. Make your sentence is in active voice (also check throughout your paper). Who did it (somebody else, not you)?

Introduction:

L31-33, unclear sentence. Please rewrite. Also, L33, “in the last few decades…..”?

L34, don't use this or that. what is “this” referring here? Try to avoid too much demonstratives in your sentences. As antecedents are not always clear, it’s more proactive to spell out the mechanisms. Also, see L38, “from this perspective” …and more………

L34-37, unclear sentence.

L41-43, for the importance of weed flora, refer: 1. https://www.sciencedirect.com/science/article/pii/S016788091830433X 2. Jordan, N., Vatovec, C., 2004. Agroecological benefits from weeds. In: Weed Ecology and Management, Iderjit (ed.), Kluwer Academic Publishers, Netherlands. 3. https://link.springer.com/article/10.1007/s13593-015-0302-5

L54, “…potential differentiation of…”… Not clear.

L65-68, unclear sentence.

L69, …”which was studied……” who studied? need active voice.

L74-75, this sentence should go with the following. Please check throughout the paper that you have paragraphs of a single sentence in multiple places, which are not justifiable (for e.g., L89, L103-108…..and so on).

L 79, “In the light of the above” is unnecessary.

Methods: 

L86, replace “analyzed by “assessed”,. Still needs to be in active voice.

L113, what do you mean by “experimental devices”?

L129, why do you need two diversity indices? Not justifiable. Also, did you try if diversity and evenness were correlated? If so, why do you need both metrices: diversity and evenness?

L134, unclear. why arcsine? See, https://esajournals.onlinelibrary.wiley.com/doi/full/10.1890/10-0340.1

L136-139, not clear. did you use ANOVA before Mixed models? why? Based on your results, I think you had year as fixed too. Also, incomplete sentence

L141, never talked about “PCA” before. Also, the use of PCA (by itself and vs other multivariate tools) is not justified.

L150, add comma after experiment.

L152, again use active voice.

Results:

L166, Fig. 1, did you analyze separately for years? not clear in your analysis section.

L170, observed? instead of counted.

L181, table and analysis at the bottom does not make sense to me. What is the point of those p-values for each species (could you use only 5 most dominant or something like that?)? Also, why different font sizes in this table? (same for Table 5)

L193, what is “lower significant”?

L199, again, were diversity and evenness correlated?

L209, Fig.2 was not clearly justified/explained in the discussion.

L222-2223, L225-227, save these sentences for the discussion (same to L282-284). Also pay attention to sentence structure. Awkward sentence.

L232, do you need already in this sentence?

L233, what is significant? there is no such thing in science. There is either an effect or no effect! Rewrite. Also, see L260 and others.

L236, “significantly”?

L263-264, mostly useless sentence. Also, unclear.

L273-276, make concise and clear. Mostly unnecessary words.

L294, why do you have % values ion PCA plot (Fig. 4) for USN but not for OTV (Fig. 2)..should be consistent.

Discussion:

L320-321, is it unexpected at all to have different results in different sites? no!

L322, “patterns”? Please pay special attention to your word and sentence structures.

L328-330, only six, every year, every site? But you have more than six species? Not clear.

L335-336, unclear sentence.

L339, encompassed? Again, please pay special attention to your word, sentence structures, and grammars.

L364, this cannot be a separate section. Your discussion should be all integrated.

L371-377, how relevant is this to your work? Sounds like too much speculation. PCA work not justified properly, unless you want say “PCA was used to assess the community composition of weeds flora across different treatment levels…..etc.….

Conclusions:

L385, derived?

Author Response

Response to Reviewer#2

Review on sustainability-453011 “The influence of herbicide underdosage on the composition and diversity of weeds in oilseed rape Mediterranean fields”

This manuscript evaluates the effects of different herbicide treatment levels on weed floral composition of oilseed rape (Brassica napus L.?? scientific name should be included) at two different sites in Italy.

I think the method used by the authors is vaguely enough, but the statistical analysis part is not clear and strong. The power of their work could have been stronger if they had more sites than just two (particularly 0because of the high variation between sites), but perhaps the authors could address this in the methods or in the discussion, including some logistical constraints. The manuscript is not clearly written at many places and needs to be improved to justify for the publication. Also, language in several places could have been more straightforward (i.e., active voice instead of passive) and concise. For example, there is no need of explaining something like “The results of the PCA are shown in table 7 (L275)…”. Please pay special attention to maintain the standards of scientific writings. Overall, the results of this study are interesting and useful. Detailed comments are below.

Detailed comments.

Title:

The scientific name of oilseed rape was added in the title, while it was removed frm keywords.

Abstract:

L13, Influence is not the right word here.

We placed “effect” instead of “influence” (L. 13).

L16, authors need to pay attention to the sentence structure. Also, need “and” after PE. Make your sentence is in active voice (also check throughout your paper). Who did it (somebody else, not you)?

Done, the structure of the sentence has been changed in active voice, and “and” has been added after PE (LL. 14-17).

Introduction:

L31-33, unclear sentence. Please rewrite. Also, L33, “in the last few decades…..”?

We rephrased the two sentences to make them clearer to the reader (LL. 33-37).

L34, don't use this or that. what is “this” referring here? Try to avoid too much demonstratives in your sentences. As antecedents are not always clear, it’s more proactive to spell out the mechanisms. Also, see L38, “from this perspective” …and more………

Thank you for the suggestion, we deleted “this” (L. 37 and L. 40).

L34-37, unclear sentence.

We improved the readibility of the sentence (LL. 35-39).

L41-43, for the importance of weed flora, refer: 1. https://www.sciencedirect.com/science/article/pii/S016788091830433X 2. Jordan, N., Vatovec, C., 2004. Agroecological benefits from weeds. In: Weed Ecology and Management, Iderjit (ed.), Kluwer Academic Publishers, Netherlands. 3. https://link.springer.com/article/10.1007/s13593-015-0302-5

We added the suggested references (L. 45).

L54, “…potential differentiation of…”… Not clear.

We replaced the term “differentiation” with “onset” in order to make the sentence clearer (L. 57).

L65-68, unclear sentence.

We rephrased the sentence to make it clearer (LL. 74-78).

L69, …”which was studied……” who studied? need active voice.

We changed the sentence in active voice (L. 79).

L74-75, this sentence should go with the following. Please check throughout the paper that you have paragraphs of a single sentence in multiple places, which are not justifiable (for e.g., L89, L103-108…..and so on).

We removed the sentence (LL. 84-85). The sentences in line 89 and lines 103-108 were not deleted because we were asked by another reviewer to integrate them with further details (L. 99 and LL. 113-114).

L 79, “In the light of the above” is unnecessary.

We removed “In the light of the above” (L. 89).

Methods:

L86, replace “analyzed by “assessed”,. Still needs to be in active voice.

We replaced “analyzed” with “assessed”, and we rephrased the sentence in active voice (L. 96).

L113, what do you mean by “experimental devices”?

We meant experiments, we placed experiments instead of “experimental devices” (L. 125).

L129, why do you need two diversity indices? Not justifiable.

As stated by Hanzlik and Gerowitt (Agron. Sustain. Dev. (2016) 36: 11. https://doi.org/10.1007/s13593-015-0345-7) and Nkoa et al. (Weed Sci. (2015) 63:64-90. DOI: 10.1614/WS-D-13-00075.1) an advantage of using such indices is that—in contrast to abundance measures—they contain information about the entirety weed vegetation of the field. In this way Shannon and Simpson indeces were used to determine differences in the specific aspects of diversity.

Also, did you try if diversity and evenness were correlated? If so, why do you need both metrices: diversity and evenness?

We agree with the reviewer. Shannon and evenness are obviously correlated. We removed the evenness index and we modified the text of the manuscript accordingly (LL. 143-144).

L134, unclear. why arcsine? See, https://esajournals.onlinelibrary.wiley.com/doi/full/10.1890/10-0340.1

Unfortunately, there was an old sentence related to the first draft of the manuscript. We rephrased some sentences in order to make the text clearer to the reader (LL. 148-155).

L136-139, not clear. did you use ANOVA before Mixed models? why? Based on your results, I think you had year as fixed too. Also, incomplete sentence.

As stated above, we changed the sentence in order to make it clearer to the reader. The reference to ANOVA was a misprint of an older version of the manuscript draft. The year was considered as fixed effect, and the sentence was completed accordingly (LL. 150-155).

L141, never talked about “PCA” before.

We put PCA within square brackets and add Principal Component Analysis in the sentence (L. 159).

Also, the use of PCA (by itself and vs other multivariate tools) is not justified.

We added a short paragraph (in “2.4 Data processing and statistical analyses” subsection) where we justified the use of PCA (LL. 156-176).

L150, add comma after experiment.

Done (L. 177).

L152, again use active voice.

Done, we changed the sentence by using active voice (L. 177).

Results:

L166, Fig. 1, did you analyze separately for years? not clear in your analysis section.

The statistical analysis is reported separately for each year, because we found a significant year x treatment interaction. We improved the text by specifying it (LL. 153-155).

L170, observed? instead of counted.

We placed “observed” instead of “counted” (L. 198).

L181, table and analysis at the bottom does not make sense to me. What is the point of those p-values for each species (could you use only 5 most dominant or something like that?)? Also, why different font sizes in this table? (same for Table 5)

We agree with your suggestion, and we moved the less relative abundant species in a table placed in the supplementary material (Table A5). We uniformed the font sizes in Tables 2 and 5 and we modified the text of the manuscript accordingly (LL. 201-206).

L193, what is “lower significant”?

Thank you for this advice, we removed the sentence (LL. 224-225).

L199, again, were diversity and evenness correlated?

As we stated above, eveness index was removed.

L209, Fig.2 was not clearly justified/explained in the discussion.

Thank you for this advice, we included the discussion of Fig. 2 in LL. 426-430.

L222-223, L225-227, save these sentences for the discussion (same to L282-284). Also pay attention to sentence structure. Awkward sentence.

The sentences were removed.

L232, do you need already in this sentence?

We removed “already” (L. 265).

L233, what is significant? there is no such thing in science. There is either an effect or no effect! Rewrite. Also, see L260 and others.

We added “statistically” before “significant” (L. 266). The same in L. 267.

L236, “significantly”?

We placed “significantly” instead of “significant” (L. 269).

L263-264, mostly useless sentence. Also, unclear.

We deleted it (LL. 298-300).

L273-276, make concise and clear. Mostly unnecessary words.

We deleted the first two sentences, and “in our analysis” in the third one (LL. 308-310).

L294, why do you have % values ion PCA plot (Fig. 4) for USN but not for OTV (Fig. 2). should be consistent.

Thank you for this advice, we provided to uniform the two figures (L. 242).

Discussion:

L320-321, is it unexpected at all to have different results in different sites? no!

We deleted the sentence (LL. 355-356).

L322, “patterns”? Please pay special attention to your word and sentence structures.

Done, thank you (L. 357).

L328-330, only six, every year, every site? But you have more than six species? Not clear.

We meant that out of all observed species, six species were always present in each year and in each site. We rephrased the sentence (LL. 363-365).

L335-336, unclear sentence.

We rephrased the sentence (LL. 369-372).

L339, encompassed? Again, please pay special attention to your word, sentence structures, and grammars.

“Encompass as” was deleted (L. 375).

L364, this cannot be a separate section. Your discussion should be all integrated.

We delected the subsections from discussion section (L. 340).

L371-377, how relevant is this to your work? Sounds like too much speculation. PCA work not justified properly, unless you want say “PCA was used to assess the community composition of weeds flora across different treatment levels…..etc.

We add a new part to discuss the relevance of PCA results (LL. 426-435).

Conclusions:

L385, derived?

We replaced “deriving” with “derived” (L. 457).

Reviewer 3 Report

see attached comments

Author Response

Response to Reviewer#3

In the present paper, authors present the results of two three‐year experiments carried out at two different sites analysing the influence of six herbicide treatments on the weed flora composition.

The topic is interesting and actual.

We would like to thank the reviewer for this encouraging evaluation.

Some remarks here below.

Introduction:

Author say that “this study was aimed to analyze the effects of underdosed herbicides and untreated plots on floristic composition and species diversity in a Mediterranean oilseed rape cropping system.”

However looking at the previous paper from the same authors (Low-Input Herbicide Management: Effects on Rapeseed Production and Profitability), I have the impression that the aim of the project was mainly to reduce the amount of herbicides keeping oilseed rape production, rather than enhancing diversity of weeds.

The impression of the reviewer is right. The primary objective of the research project was to investigate the feasibility of reducing the use of herbicide also maintenaing a good seed production in a perspective of economical sustainability (profitability).

However, a specific objective was also to investigate the dynamic and the evolution of the weed flora composition as a consequence of herbicide reduced doses and in a perspective of environmental sustainability (maintenance of weed flora diversity).

Instead of pooling all the results in a single manuscript, resulting messy, confusing or not sufficiently detailed, all authors jointly decided that the better approach would have been to prepar two separate manuscripts, the first focused on economic aspects of sustainability (both seed production and profitability) and the second one more focused on environmental aspects of sustainability (weed flora diversity assessment).

Materials and Methods:

As a consequence of the previous point, I do not see clear reasons to change the treatments applied during the 3‐year experiment. In other words: why changing the composition during the 3 years? Trifluralin was eliminated and Clopiralid + propaquizafop was added: in such a short experiment it is not possible to understand the effect of such modifications with respect to the rest of the protocol. In other words it is not possible to understand if changes are related to the modified treatments or to the natural evolution of the weeds in the area.

Trifluralin was substitued because, with respect to the time when research project was evaluated and funded, it was banned by the European Union. Consequently taking into account also the results of the first year of trial (in both sites) we prefered to replace Trifluralin with another treatment. We also specified this issue by revising the Methods accordingly (LL. 115-116). As regard Clopiralid + propaquizafop, in the previous manuscript we explained that “…due to the poor performance of post-emergence treatment during the two previous growing cycles (2007–2008 and 2008–2009), in 2010, a combined treatment (POE) of clopiralid (Lontrel 72 SG, Dow AgroSciences Italia) + propaquizafop (Agil, DuPont Italia) active ingredients (200 g a.i. ha−1 and 100 g a.i. ha−1, respectively) was applied….”. Before drafting the present paper, we statistically processed the data and we didn’t find any significant effect of post-emergence treatment on weed flora richness and diversity between first year (post-emergence treatment with metazachlor) and last year (post-emergence treatment with clopiralid). As consequence we decided to include this treatment in the present paper on weed flora composition, discussing more on the time of application (e.g. LL. 452-455) that on the type of active ingredient.

Total rainfall in 2008–2009 and 2009–2010 was 51% and 38% higher than the historical averages; also temperatures were quite high during the first eight weeks of crop cycle in 2009-2010. Authors should discuss in a deeper way (much more than lines 349-356) the effects that such specific weather conditions had on weeds growth, otherwise it is not clear if weeds evolution is related to weather or to doses.

Thank you for your suggestion, we included a paragraph that highlighs the importance of weather conditions in the discussion section (LL. 398-411).

The “weedy treatment” is not clear and should be better described and specified.

Thank you for your suggestion, we improved the Material and Methods (LL. 113-114), by adding details about the weedy treatment.

Results:

As discussed in their previous paper, authors made analysis also in a control condition, without treatments. The actual weed variability improvement allowed by the reduced doses has to be done against the ideal condition (= no treatment) which should help to define the reference ideal case.

The authors aware that a statistical test such as the Dunnett one might be suitable for testing herbicide treatments against the weedy treatment. However, the Dunnett test has the disadvantage that it does not compare the treatments other than the weedy treatment among themselves at all (Lee and Lee, 2018 doi: 10.4097/kja.d.18.00242).

In the case of the present paper, the aim was to analyze the difference among treatments, in order to investigate the weed flora compositions along a treatment gradient from the unsprayed check to the labelled treated one as a relationship between treatments and weed flora composition.

Thus, our approach, taking into account recent published studies dealing with herbicide treatment trials (e.g Bajwa et al., 2019 https://doi.org/10.1016/j.cropro.2018.11.009; Sing et al., 2018 https://doi.org/10.1016/j.fcr.2018.03.002), is consistent with the aim of the paper and we would prefer to not address this reviewer’s suggestion.

Results should then be presented on the basis of such ideal case. On the other hand, reported percentages (Table 2 and Table 5) give only a confused idea on the actual variability improvements.

Thank you for the suggestion, but as we mentioned earlier, the aim was to provide a comprehensive description of the weed flora composition under reduced herbicide doses. We think that presentation of the results taking into account weed abundance is an added valuable information in drawing weed flora composition and floristic structure and it is consistent with what Hanzlik and Gerowitt (Agron. Sustain. Dev. (2016) 36: 11. https://doi.org/10.1007/s13593-015-0345-7) stated in their review on methods to conduct and analyse weed surveys.

Tables 2 and 5 report relative weed abundance percentage in order to highlight the measurement of the occurrence of weed species in the overall weed plant community and within each treatment, rather than underline actual variability improvements. In our opinion, and consistently with prior researches, diversity indices, such as Shannon index, are more suitable to display the variability among treatments, and within treatment, among growing seasons (Tables 3 and 6).

Round 2

Reviewer 2 Report

Abstract:

L13, could you add a background (justification of your study) sentence at the beginning of your abstract?

L17, you used the term “weed floral composition”. You also used a term “assemblage” in L18. Are they different? How?

L20, what is “labelled treated”? I understand it but, make it more clear for others.

L21, replace “proved to be” by “were”.

L21, what do you mean by “under in”?

L23, “achieved” or “observed”?

L24, what is “slight similar”? Did you mean slightly? Please pay close attention to the word choice, and sentence and grammatical structures throughout the text.

Introduction:

L30-31, please revise the sentence. For example, what is key-point? Why do you need both agroecosystem and cropping systems in this sentence?

I may not have any comments below (because it is very common) on sentence structures and grammars, BUT please be careful to pay special attention on this.

L39, “player” is a colloquial word. Please replace by a more scientific word.

L96-98, the word “cropping’ is repeated three times. Is it necessary?

L99, “factors”, not “factor” ……Again, this is just an example, you need to pay close attention on words and grammar.

Methods:

L117 (also L751), just composition? You said “structure and assemblage” too?

L120 and elsewhere. I commented this in earlier version too that don’t use a single-sentenced paragraph very often, unless there is a good reason. Also see L33-34; L35-36, L940-942, and elsewhere.

L127, distributed or applied?

L346-352, L354, L367, and elsewhere, I already told you in the first round of comments that you should use an active voice predominantly.

L440, analyzed or processed?

Results:

L546, weren’t Shannon and Simpson diversity indices correlated?

L560-562, as I stressed this before, you can include these tables or figures in the text; in that situation, these sentences seem unnecessary.

L594 and elsewhere, I am still not convinced that you need to use the terms such as “significant” or statistically. These words are implied when you say “there is a difference” ….Or, we don’t use the word “different” if your treatments are not “statistically significant”…Does this make sense?

For example, see L617; this is what I meant.

Discussion:

L696, what is sustainable intensification? Clarify.

L707, slightly?

L768, L771, L772 C. album should be italics.

Conclusion:

L976-979, revise your sentence or breakdown into two. You last sentence should be more clear.

Author Response

Response to reviewer#2

Abstract:

L13, could you add a background (justification of your study) sentence at the beginning of your abstract?

We added a sentence in the abstract to justify the study (LL. 13-14). However, we deleted some words (e.g. L. 18, LL. 19-20) in order to stay within the available 200 words. 

L17, you used the term “weed floral composition”. You also used a term “assemblage” in L18. Are they different? How?

We used these terms as synonyms, since we were asked by the editorial revision to change some words or sentences that overlapped to previous published papers (see also Tang et al., 2013 doi:10.1093/jpe/rtt018, where the terms are used as synonyms).

L20, what is “labelled treated”? I understand it but, make it more clear for others.

We placed “M100 treated” instead of “labelled treated” (L. 24).

L21, replace “proved to be” by “were”.

Done (L. 25).

L21, what do you mean by “under in”?

Thank you for this advice. We deleted “under” (L. 25).

L23, “achieved” or “observed”?

We replaced “achieved” with “observed” (L. 27).

L24, what is “slight similar”? Did you mean slightly? Please pay close attention to the word choice, and sentence and grammatical structures throughout the text.

We replaced “slight” with “slightly” (L. 28).

Introduction:

L30-31, please revise the sentence. For example, what is key-point? Why do you need both agroecosystem and cropping systems in this sentence?

“Key-point” means that biodiversity plays a crucial role at agricultural landscape level, we changed the sentence, accordingly (LL. 35-37). We used both agroecosystem and cropping systems because with agroecosystem we meant the agricultural landscape in which the cropping system occurs and interacts with (according also with the cited references). We placed agricultural landscape in place of agroecosystem.

I may not have any comments below (because it is very common) on sentence structures and grammars, BUT please be careful to pay special attention on this.

Thank you for this advice, we will request for MDPI English language editing service.

L39, “player” is a colloquial word. Please replace by a more scientific word.

We deleted “player” and placed “fundamental”.

L96-98, the word “cropping’ is repeated three times. Is it necessary?

Thank you for this advice, we deleted two of the three “cropping” (LL. 75-76).

L99, “factors”, not “factor” ……Again, this is just an example, you need to pay close attention on words and grammar.

Done (LL. 77).

Methods:

L117 (also L751), just composition? You said “structure and assemblage” too?

We added structure and assemblage between round brackets (L. 98).

L120 and elsewhere. I commented this in earlier version too that don’t use a single-sentenced paragraph very often, unless there is a good reason. Also see L33-34; L35-36, L940-942, and elsewhere.

L. 120 was deleted. As regards L. 33-34 and L. 35-36 we would prefer to not address the reviewer’s suggestion, because those sentences contain relevant references that allow us to introduce the background in which the paper develops. L 940-942 was removed.

L127, distributed or applied?

As explained above, some words or sentences were changed to accomplish editorial revision.

L346-352, L354, L367, and elsewhere, I already told you in the first round of comments that you should use an active voice predominantly.

Done, LL. 129-139.

L440, analyzed or processed?

We placed analyzed instead of processed (L. 154).

Results:

L546, weren’t Shannon and Simpson diversity indices correlated?

Yes, they are correlated, but the existence of correlations between diversity measures should not be surprising as they represent aspects of the same phenomenon (as also claimed Morris et al., 2014 doi: 10.1002/ece3.1155).

We stated in the round-1 of revision that the use of both diversity indices improves the output information of the dataset, which is unique for each community or sample analyzed. Looking at the wider content, the use of both indices in parallel adds a more complex information of the diversity in the studied system. In this sense, Shannon index has an advantage over Simpson because it depends more on species richness and less abundant species, so it is very sensitive to even small diversity changes, and thus is widely used to assess the actual state of the studied system. On the other hand, Simpson index has the advantage over Shannon index in counting more on dominant species and is not affected by less abundant elements.

L560-562, as I stressed this before, you can include these tables or figures in the text; in that situation, these sentences seem unnecessary.

We deleted the sentence (LL. 244-245).

L594 and elsewhere, I am still not convinced that you need to use the terms such as “significant” or statistically. These words are implied when you say “there is a difference” ….Or, we don’t use the word “different” if your treatments are not “statistically significant”…Does this make sense?

For example, see L617; this is what I meant.

We agree. We reformulated the following paragraphs: LL. 193-198, LL. 273-279, LL. 305-307.

Discussion:

L696, what is sustainable intensification? Clarify.

We don’t explained it in the text, because we cited two papers (Albrecht  et al., 2016 DOI:10.1080/23818107.2016.1237886; Petit et al., 2018 DOI:10.1007/s13593-018-0525-3) where the concept of sustainable intensification was clearly explained. Sustainable intensification is related to the FAO policy that promotes practices to produce more per unit of input (or maintaining production with less input).

L707, slightly?

Done, we replaced “slight” with “slightly”.

L768, L771, L772 C. album should be italics.

Done.

Conclusion:

L976-979, revise your sentence or breakdown into two. You last sentence should be more clear.

We shortened the sentence to make it clearer (LL. 468-471).

Reviewer 3 Report

I appreciate the efforts of the authors to improve the quality of the paper. I still believe table representation in the last tables could be improved, however the paper is now acceptable for publication.

Author Response

Response to reviewer#3

I appreciate the efforts of the authors to improve the quality of the paper. I still believe table representation in the last tables could be improved, however the paper is now acceptable for publication.

Thank you for the positive feedback. To address your concern, in the new version of manuscript, we improved tables 2 and 5. We added a row to separate years and analysis of variance, we increased the font size, and the right and left margins of each cell.